# Characterizing human mobility patterns in rural settings of sub-Saharan Africa

Hannah R Meredith[1]*, John R Giles[1], Javier Perez-Saez[1], Théophile Mande[2], Andrea Rinaldo[3,4], Simon Mutembo[5,6], Elliot N Kabalo[7], Kabondo Makungo[8], Caroline O Buckee[9], Andrew J Tatem[10], C Jessica E Metcalf[11], Amy Wesolowski[1]*

[1]Department of Epidemiology, Johns Hopkins Bloomberg School of Public Health, Baltimore, United States; [2]Bureau d'Etudes Scientifiques et Techniques - Eau, Energie, Environnement (BEST-3E), Ouagadougou, Burkina Faso; [3]Dipartimento di Ingegneria Civile Edile ed Ambientale, Università di Padova, Padova, Italy; [4]Laboratory of Ecohydrology, School of Architecture, Civil and Environmental Engineering, École Polytechnique Fédérale de Lausanne, Lausanne, Switzerland; [5]Department of International Health, International Vaccine Access Center, Johns Hopkins Bloomberg School of Public Health, Baltimore, United States; [6]Macha Research Trust, Choma, Zambia; [7]Zambia Information and Communications Technology Authority, Lusaka, Zambia; [8]Zamtel, Lusaka, Zambia; [9]Department of Epidemiology and the Center for Communicable Disease Dynamics, Harvard TH Chan School of Public Health, Boston, United States; [10]WorldPop, School of Geography and Environmental Science, University of Southampton, Southampton, United Kingdom; [11]Department of Ecology and Evolutionary Biology and the Princeton School of Public and International Affairs, Princeton University, Princeton, United States

*For correspondence:
hmeredi4@jhu.edu (HRM);
awesolowski@jhu.edu (AW)

**Abstract** Human mobility is a core component of human behavior and its quantification is critical for understanding its impact on infectious disease transmission, traffic forecasting, access to resources and care, intervention strategies, and migratory flows. When mobility data are limited, spatial interaction models have been widely used to estimate human travel, but have not been extensively validated in low- and middle-income settings. Geographic, sociodemographic, and infrastructure differences may impact the ability for models to capture these patterns, particularly in rural settings. Here, we analyzed mobility patterns inferred from mobile phone data in four Sub-Saharan African countries to investigate the ability for variants on gravity and radiation models to estimate travel. Adjusting the gravity model such that parameters were fit to different trip types, including travel between more or less populated areas and/or different regions, improved model fit in all four countries. This suggests that alternative models may be more useful in these settings and better able to capture the range of mobility patterns observed.

## Introduction

Human mobility patterns are a reflection of behaviors, ranging from routine (e.g., commuting daily for work or school, traveling for holidays and religious gatherings, or seeking seasonal work opportunities) to irregular (e.g., relocating due to environmental changes or crises or social distancing due to a pandemic) (*Charaudeau et al., 2014*; *Haberfeld et al., 1999*; *International Organization for Migration, 2019*; *Lessler et al., 2014*; *Lu et al., 2012*; *Pullano et al., 2020*). Consequently, characterizing human mobility patterns is important for a wide range of applications, including predicting the spread of infectious diseases, traffic forecasting, designing intervention strategies, assessing health service accessibility, planning natural disaster relief efforts, and estimating migratory flows to estimate

changes in population demographics (*Charaudeau et al., 2014*; *Dotse-Gborgbortsi et al., 2020*; *Findlater and Bogoch, 2018*; *Finger et al., 2016*; *Gilbert et al., 2020*; *Lu et al., 2012*; *Mari et al., 2017*; *Palchykov et al., 2014*; *Stoddard et al., 2009*; *Wesolowski et al., 2012*). Depending on the spatial and temporal resolution of travel relevant to the question of interest, there are many data sources that can quantify travel, such as national censuses, traffic or commuting data, travel surveys, or mobile phone data (*Tatem, 2014*). However, when mobility data are limited or unavailable, spatial interaction models are often used to estimate mobility patterns. One of the most commonly used models is the gravity model (*Wesolowski et al., 2015b*), which assumes the number of trips between an origin and destination over a fixed time period will increase as a function of the destination and origin population sizes and decrease with distance. While the gravity model was originally developed to describe commuting in a high-income setting (*Zipf, 1946*), it has been used to model travel in low- and middle-income countries (LMICs) when data are limited or unavailable (*Stone et al., 2019*; *Wells et al., 2019*; *Wesolowski et al., 2015b*). In high-income settings, the standard gravity model has been shown to perform well when predicting commuter movement between cities (*Masucci et al., 2013*) and perform poorly when predicting movement across areas with heterogeneity in demographics and population density or in rural areas (*Truscott and Ferguson, 2012*; *Xia et al., 2004*). However, the degree to which geographic, sociodemographic, and infrastructure differences may impact the ability for models to capture travel patterns in LMICs, particularly in more rural settings, remains to be determined.

Increasing mobile phone ownership (over 67 % of the global population in 2019) and the mobility patterns that can be extracted from these data provide a valuable resource for evaluating how well spatial interaction models can capture human mobility patterns in low- and middle-income settings (*GSM Association, 2020*; *Wesolowski et al., 2016*). However, the challenge of procuring mobile phone datasets has resulted in relatively few studies that fit mobility models to relevant travel data from LMICs. Of these, a few studies have explored model adjustments and show that predictions can be improved by adjusting the gravity model to account for factors such as education levels, economic opportunities, gender, environmental factors, trip duration, contiguity of origin and destination, and proportion of population living in urban areas (*Garcia et al., 2015*; *Henry et al., 2003*; *Wesolowski et al., 2015b*). Typically, studies assume that these factors impact all trips homogeneously, thus fitting a single set of parameters for all trips is a sufficient adjustment. Yet, even with such adjustments, gravity models may still fail to accurately capture mobility patterns in LMICs, particularly in rural areas (*Henry et al., 2003*; *Wesolowski et al., 2015b*), suggesting that models which account for regional heterogeneity in travel may improve their estimates in these settings.

Here, we examine the impact that regional heterogeneity and urbanicity has on travel patterns as well as how well spatial interaction models can reproduce travel patterns in LMICs by examining four countries in Sub-Saharan Africa: Namibia, Kenya, Burkina Faso, and Zambia (*Figure 1A–D*). These countries capture a range of levels of urbanization, population distribution, and importantly include large rural areas common in many LMICs. Using mobile phone data, we first characterized human mobility patterns from these four countries and determined which features were not well captured by basic gravity models. Next, we conducted an analysis of six variations of the basic gravity model, including variations that account for regional travel and urbanicity, as well as a basic radiation model to determine which provided the best trip estimates for each country (*Simini et al., 2012*). Finally, we compared the different model fits across the four countries to evaluate if the adjustments produced similar improvements for all countries. To our knowledge, no other models have captured these mobility features, been tested consistently against mobility data from multiple LMICs, and provided clear guidance on which model to use and when. This study provides further insight on the mobility patterns in LMICs and highlights where mobility model estimates may deviate when applied in other similar settings, ultimately improving our understanding of topics such as disease spread, migratory flows, and intervention efficacy in LMICs.

## Results

We analyzed the average number of daily trips, defined here as the number of subscribers moving from one district (administrative level 2) to another, per month extracted from each mobile phone dataset (see Materials and methods). Similar to other studies, we found that the average number of monthly

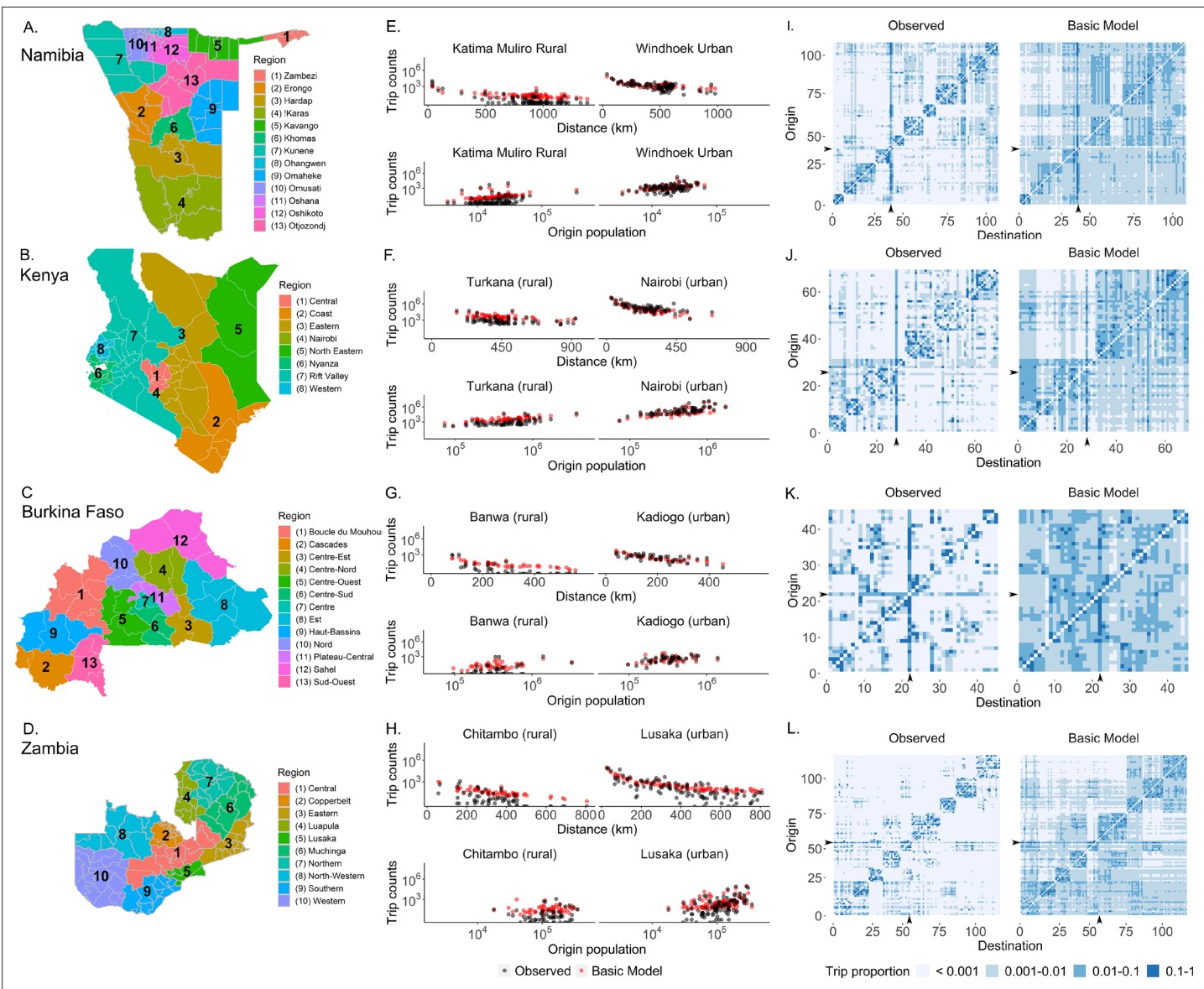

**Figure 1.** The mobility patterns extracted from mobile phone data from four countries in Sub-Saharan Africa. (**A–D**) Data from four Sub-Saharan African countries were selected to characterize human mobility patterns: Namibia, Kenya, Burkina Faso, and Zambia. Travel between districts (administrative level 2) was estimated via mobile phone data from each country. A basic gravity model was fit to trip data from each country which assumes that the number of trips decreases with distance and increases with population size (**E–H**). Here, one rural (left panel) and one urban (right panel) destination were selected from each country to show that, while the observed trips (black) from different origins do generally follow the assumptions of the gravity model (red), the gravity model is not fully capturing the observed trip patterns. See *Figure 1—figure supplements 1–8* for this comparison made for all districts in each country. (**I–L**) Comparisons of origin-destination matrices colored by trip proportions estimated by the basic gravity model and the mobile phone data (observed) highlight how the basic gravity model tends to overestimate many trips, particularly those that are off-diagonal (e.g., inter-regional trips). The columns and rows of the OD matrix are ordered by district ID, which were assigned such that districts within the same region (adminstrative level 1) were clustered together. The capital district is indicated by the black arrow on the x- and y-axes. The colors indicate the proportion of an origin's trips made to each destination (with light blue representing destinations visited infrequently and dark blue representing destinations visited most frequently). (See *Supplementary file 1B* for the key to the origin and destination numbers and *Figure 1—figure supplement 9* for district level maps).

The online version of this article includes the following figure supplement(s) for figure 1:

**Figure supplement 1.** Trip estimates as a function of trip distance for all districts in Namibia.

**Figure supplement 2.** Trip estimates as a function of origin population for all districts in Namibia.

**Figure supplement 3.** Trip estimates as a function of trip distance for all districts in Kenya.

*Figure 1 continued on next page*

**Figure supplement 9.** Relevant maps for Namibia, Kenya, Burkina Faso, and Zambia.

trips between districts from Namibia, Kenya, Burkina Faso, and Zambia generally decreased with trip distance and there were more trips from and to more populated areas (*Figure 1E–H*, *Figure 1— figure supplements 1–8*). Trips were concentrated between districts within the same region (administrative level 1) to varying degrees (intra-regional trips made up 30 % of all trips in Burkina Faso, 45 % in Kenya, 62 % in Namibia, and 72 % in Zambia) and to a few common destinations, including the district where the capital was located (*Figure 1I–L*, *Supplementary file 1A*). Although Namibia, Burkina Faso, and Zambia each consisted of ~95 % predominantly rural districts, the distribution of monthly trips between urban and rural districts varied across countries. The majority of Namibia's and Burkina Faso's trips were between rural locations (62% and 70.5% of all trips, respectively), while Zambia's trips were split between rural locations (53%) or rural and urban locations (46%). Kenya, with 56 % predominantly urban districts, had the largest proportion of monthly trips between urban locations (70%). As a basic model, we first estimated trip counts using a basic gravity model (single parameter set fitted for all trips), which is based on the population sizes of the origin and destination and the distance between locations, with two variants on the distance kernel (power or exponential decay) (Equation 1). Estimates from this basic model overestimated trip counts and missed important features of the data, such as higher trip counts in short-distance and within-region trips, relative to

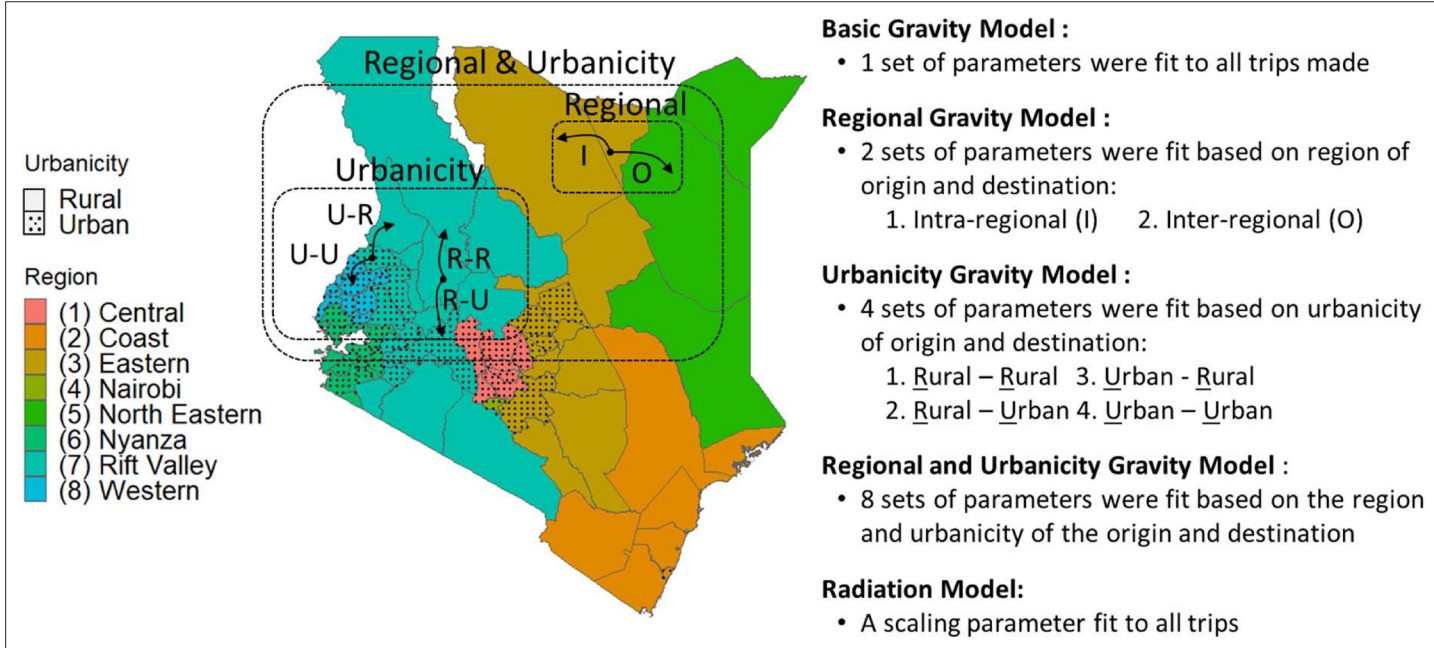

**Figure 2.** Variations of the gravity model were fit to data to capture different types of trips. Here, Kenya is used to demonstrate the trip types that could be defined by the region (represented by color) and/or urbanicity (solid = predominantly rural, dotted = predominantly urban) of the trip's origin and destination to outline the various models fit. See *Figure 2—figure supplements 1–2* for the fitted parameters.

The online version of this article includes the following source data and figure supplement(s) for figure 2:

**Source data 1.** Table of model parameter values fit for each country, both distance kernels, and all trip types.

**Figure supplement 1.** Gravity model parameters (power distance kernel).

**Figure supplement 2.** Gravity model parameters (exponential distance kernel).

**Table 1.** Gravity model variations (using power distance kernel) and radiation model, ranked for each country based on Deviance Information Criterion (DIC) and percent change (%Δ) from the basic gravity model.
A similar trend was seen for gravity models with exponential distance kernel (*Supplementary file 2A*).

| Rank | Namibia | | Kenya | | Burkina Faso | | Zambia | |
|---|---|---|---|---|---|---|---|---|
| | Model | DIC (%Δ) | Model | DIC (%Δ) | Model | DIC (%Δ) | Model | DIC (%Δ) |
| 1 | Reg - Urb. | 3.62E + 06 (41.0) | Reg - Urb. | 2.43E + 08 (30.1) | Reg - Urb. | 1.93E + 05 (27.7) | Reg - Urb. | 2.01E + 06 (16.3) |
| 2 | Urbanicity | 4.56E + 06 (25.6) | Urbanicity | 2.53E + 08 (27.2) | Urbanicity | 2.05E + 05 (23.1) | Regional | 2.08E + 06 (13.4) |
| 3 | Regional | 4.61E + 06 (24.8) | Regional | 3.40E + 08 (2.1) | Regional | 2.52E + 05 (5.7) | Urbanicity | 2.38E + 06 (1.1) |
| 4 | Basic | 6.12E + 06 (0.0) | Basic | 3.48E + 08 (0.0) | Basic | 2.67E + 05 (0.0) | Basic | 2.40E + 06 (0.0) |
| 5 | Radiation | 8.68E + 06 (-41.7) | Radiation | 4.26E + 08 (-22.4) | Radiation | 3.39E + 05 (-27.2) | Radiation | 4.3E + 06 (-79.9) |

long-distance and between-region trips (*Figure 1E–L*, *Figure 1—figure supplements 1–8*). Travel involving predominantly rural locations tended to be overestimated and these results were largely observed for both distance kernels and for all countries.

Given how the basic model's estimates of urbanicity and regional movement deviated from the call data records, we tested six additional variations of a gravity model that allowed for parameters to capture increasingly complex features in the data (Equations 2–9). These included fitting parameters for the origin and destination population sizes and trip distance to: (1) trips defined by origin and destination population density (urbanicity model: higher population density = predominantly urban, lower population density = predominantly rural), (2) trips within- and between-regions (regional model), and (3) trips defined by both region and urbanicity (regional-urbanicity model) (*Figure 2*). Each model variation was tested with two distance kernels (power and exponential decay). We also evaluated a basic radiation model (Equation 11; *Masucci et al., 2013*; *Simini et al., 2012*).

All parameterized gravity models outperformed the basic model. Allowing for parameters to be fit by urbanicity or region improved model fit to varying degrees, depending on the country (*Table 1*, *Supplementary file 2A*). Accounting for urbanicity provided a larger improvement in model fit than region for all countries; however, this improvement was larger in Kenya and Burkina Faso than Namibia and was only observed for Zambia if the lower urbanicity threshold (10%) was implemented. In these models, the degree to which the parameters varied increased with model complexity, with the distance parameter varying the most, allowing for different weights to be applied based on the trip type (*Figure 2—figure supplements 1–2*). Overall, the most complex model, the regional-urbanicity model, had the best fit for all countries (the power variant reduced the basic model's deviance information criterion (DIC) by 41 % for Namibia, 30 % for Kenya, 28 % for Burkina Faso, and 16 % for Zambia) (*Table 1*, *Supplementary file 2A*). Its flexibility allowed for the importance of origin and destination population sizes to vary by trip type, distinguished differences in the relative importance of distance by trip type, was able to better distinguish between inter- and intra-regional trip estimates, and adjusted for the lower trip counts between rural locations, relative to other trip types (*Figure 3A*, *Figure 3—figure supplements 1–4*). This was true regardless of the distance decay function assumed, although the best fitting decay function did vary by country (*Supplementary file 2A*). Interestingly, the radiation model had the most variable performance of all models, where it was the poorest performing in Namibia, Burkina Faso, and Zambia, but outperformed the basic gravity model (exponential form) in Kenya. Differences in the population distribution (more heterogenous in Namibia, Burkina Faso, and Zambia than Kenya) and locations of more populated areas may be the cause of these conflicting findings across countries (*Figure 1—figure supplement 9*; *Linard et al., 2012*; *Masucci et al., 2013*). These results were consistent across a range of administrative levels (1-3), suggesting that including locations' urbanicity and/or region in a model will generally improve model fit across spatial scales (*Supplementary file 2B*).

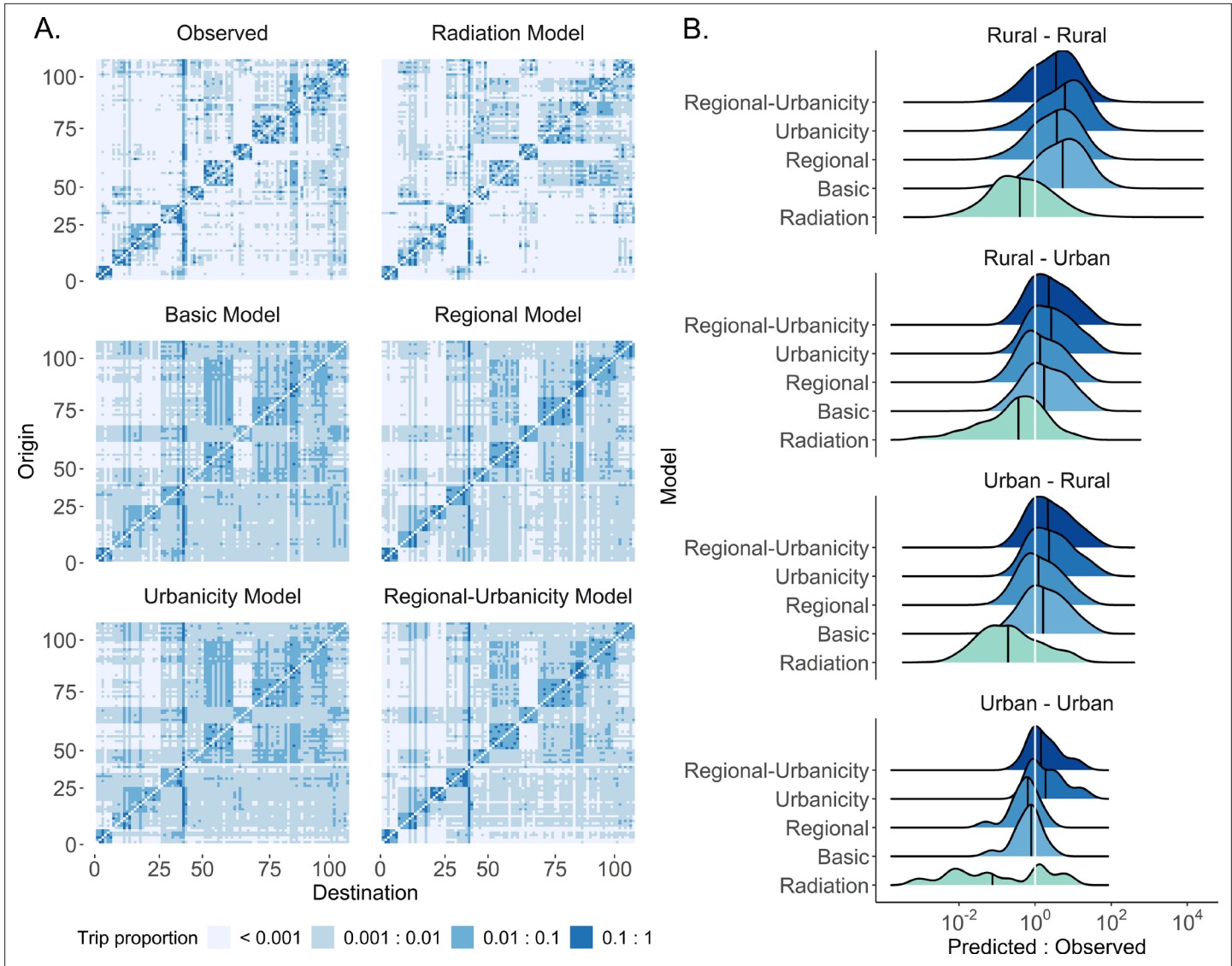

**Figure 3.** The modeled estimates of trip counties and fits by type of trip. (**A**) The proportion of trips from each origin location in Namibia was estimated by five different spatial models (power distance kernel displayed here) and ordered by the origin and destination ID. Regional clustering was more pronounced and there were fewer inter-regional trips in the adjusted models and radiation model, relative to the basic model. See *Figure 3—figure supplements 1–4* for all countries and both distance kernel functions. The columns and rows of the OD matrix are ordered by district ID. (**B**) The ratio of predicted to observed trip counts in Namibia was calculated to determine the distribution of trips that were over- (ratio >1) or underestimated (ratio <1) in each trip type by model. The median ratio (solid black vertical line) for each trip type is compared with the equity line (white vertical line) for each model (shown as different colors). The proportion of trips that fall within the selected interval (± 10 % of the observed trip count) was also used to assess a model's ability to capture trips in that category. Generally, the basic model captured urban-to-urban trips the best and overestimated the other trip types. See *Figure 3—figure supplements 5–7* for all countries, trip types, and distance kernel functions.

The online version of this article includes the following figure supplement(s) for figure 3:

**Figure supplement 1.** Predicted trip proportions for different models compared to the observed trip proportions from mobile phone data in Namibia.

**Figure supplement 2.** Predicted trip proportions for different models compared to the observed trip proportions from mobile phone data in Kenya.

**Figure supplement 3.** Predicted trip proportions for different models compared to the observed trip proportions from mobile phone data in Burkina Faso.

**Figure supplement 4.** Predicted trip proportions for different models compared to the observed trip proportions from mobile phone data in Zambia.

**Figure supplement 5.** Distribution of predicted to observed trip count ratios for each urbanicity trip category in each country.

**Figure supplement 6.** Distribution of predicted to observed trip count ratios for each regional trip category in each country.

**Figure supplement 7.** Distribution of predicted to observed trip count ratios for each regional-urbanicity trip category in each country.

Overall model fit may not accurately describe how well each model can estimate certain trip types which may be relevant for particular questions. We further evaluated how close model estimates were to the observed trip count using the ratio of predicted to observed trip counts (*Figure 3B*, *Figure 3—figure supplements 5–7*). Unsurprisingly, the regional-urbanicity model (with either the exponential or power decay) produced some of the most accurate model estimates ( ± 10 % of the observed trip counts) for most trip types in Namibia (11/14 trip categories), Kenya (7/14 trip categories), and Burkina Faso (6/9 trip categories) (*Supplementary file 3A*); however, the regional model produced the most accurate estimates for a range of trips in Zambia (5/13). The proportion of trips estimated within the 10 % margin of error was lowest for Kenya, with the most accurate models only well estimating 3–7% of trips, and highest for Namibia, with some models estimating >30% of certain trip types. Depending on the type of trip and country, simpler models may provide more accurate trip estimates than the more complex regional-urbanicity model. We found that just accounting for region may be sufficient for best capturing rural-to-rural as well as general or specific inter-regional travel in Namibia. Further, a wide range of trip types was well estimated by the radiation model in Kenya and by a basic model (exponential decay) in Zambia.

## Discussion

To date, there have been limited evaluations of mobility patterns in Sub-Saharan African countries and how the geographic, demographic, and economic differences in many LMICs impact the validity of modeling assumptions to estimate travel. By comparing and evaluating a range of gravity models in four Sub-Saharan African countries, we identified clear patterns of travel that were not well approximated using a basic gravity model. By allowing these models to accommodate differences in travel between predominantly rural and/or urban areas and within- versus between-regions, we found that the best fitting model allowed for the greatest flexibility for estimating different trip types. However, there were differences in which model was best able to estimate particular types of travel, the importance of distance and population, and the overall fit of all models to the data by country. Therefore, selecting a model that adjusts for the trip type and context of interest is important for improving estimates.

If a given application is focused on estimating specific trip types (e.g., urban residents traveling to rural areas for work), these findings may be informative for model selection in similar settings. If a country is sparsely populated with a single predominantly urban district (e.g., capital district), like Burkina Faso, then the regional-urbanicity model would be recommended, as it maximizes the proportion of trips reasonably estimated for most trip types. If specifically considering trips between predominantly rural and urban locations in a sparsely populated country with a few predominantly urban districts, like Namibia, then the regional model would provide the best estimates in these settings. Alternatively, if a country is more homogeneously populated, like Kenya, the radiation model may provide the best estimates for most specific trip types. If data are limited or unavailable, then the parameters fit to these four exemplar countries may serve as a proxy.

Moving forward, the need to select a single model could be mitigated and estimates of mobility patterns could be improved by developing an ensemble model. Recently, an ensemble model outperformed individual models in estimating human mobility patterns in Australia, combining different mobility models as well as data types to optimize mobility estimates across different spatial scales (*McCulloch et al., 2021*). Using this approach, additional individual level information (e.g., gender, age, occupation) collected from other data sources could be incorporated to study their impact on model fit. Future work could also evaluate the models tested here in other countries, both within and outside of Sub-Saharan Africa. Model estimates of travel patterns in high-income countries may also benefit from accounting for urbanicity and regionality (*Truscott and Ferguson, 2012*; *Xia et al., 2004*). The increasing availability of mobility data has enabled the comparison of global human mobility patterns and revealed that (a) longer distance trip ( > 20 km) patterns are similar across low- and high-income settings (*Kraemer et al., 2020*) and (b) that mobility patterns in rural (sparsely populated) areas differ from those in urban (densely populated) areas (*Liu et al., 2015*). This suggests that our findings based on trips aggregated to the region or district level will be generalizable to other countries from a range of income settings. However, mobility patterns at smaller spatial scales appear to be different for low- and high-income settings (*Kraemer et al., 2020*), and requires further investigation.

There are a number of caveats to be noted in this study. Mobile phone data has inherent owner, user, and coverage biases, especially in LMICs where mobility data tend to be concentrated around urban areas and roads (*Kraemer et al., 2020*; *Wesolowski et al., 2013*). Furthermore, the dataset from Burkina Faso was limited to 100,000 randomly selected subscribers (1.4 % of Burkina Faso's Telecel Faso subscribers). This smaller sample size may have resulted in the models overestimating routes that were missed in the actual dataset due to relatively small trip counts. Regardless, mobile phone data remain one of the most direct ways to gather information on a large population in LMICs. Alternative human mobility data sources that have recently become more available, such as Google Mobility or Facebook Data for Good (*Kissler et al., 2020*; *Ojal, 2020*), could verify these observations in future studies; however, smart phones are also associated with ownership biases and are currently less pervasive than general mobile phone ownership (*GSM Association, 2020*). It should be noted that models parameterized by different mobility data sources have produced different outcomes (*Panigutti et al., 2017*). Thus, the generalizability of the results ascertained from these four LMICs should continue to be evaluated and expanded upon as more mobility datasets from other settings become available. Similarly, the ability of other spatial interaction models to better estimate human mobility should be explored. We opted to focus on the gravity model because it has been the most commonly used spatial interactions model; however, one of its shortfalls is that it does not address the competition or synergism that often occurs between potential destinations (*Bjørnstad et al., 2019*). While this is addressed by the radiation model, the degree to which accounting for region and urbanicity can improve its estimates remains to be considered in the future (*Bjørnstad et al., 2019*). Another shortfall of the gravity and radiation models is their inability to capture the temporally dynamic nature of human mobility. Here, we focused on one definition and measurement of human mobility patterns; however, seasonal mobility patterns, such as those driven by seasonal work or societal factors (*Buckee et al., 2017*; *Wesolowski et al., 2015a*), and irregular migration, such as those driven by political or natural crises and environmental changes (*International Organization for Migration, 2019*), are ubiquitous. Mobile phone data could also be used to quantify these movements and improve estimates of population distribution and movement over time (*Facebook data for good, 2021*; *Finger et al., 2016*; *Wesolowski et al., 2015a*). Finally, the modifiable areal unit problem (MAUP) is a source of bias in the spatial distribution and aggregation of both CDRs (and other movement data) and population that impacts the results of models dependent on these inputs (*Fotheringham and Wong, 2016*). Although the general trends in model fits were preserved across a range of administrative levels, the generalization of the results to movement aggregated at finer spatial scales or for different geographical boundaries remains to be determined.

Ultimately, gaining a more complete understanding of travel in diverse geographies will help inform applications in the health, economic, social, and transportation sciences. While incorporating urbanicity and region did improve the gravity model fit to various degrees for different countries, this study highlights the need to continue honing a model framework that can better capture mobility patterns and behavioral nuances in LMICs.

## Materials and methods

### Population, urbanicity, and geolocation data

WorldPop population data for each country were analyzed as people per pixel for each district (https://www.worldpop.org/) (*Supplementary file 1A*). WorldPop gridded building pattern datasets were used to categorize grid cells of each district as urban or rural as described elsewhere (*Dooley et al., 2020*) using QGIS v3.6. Districts with more or less than 50 % urban grid cells were categorized as urban and rural, respectively. A sensitivity analysis in which urban thresholds of 10% and 50% urban grid cells were compared showed that, while the general trends in model fits remained the same, the overall model fits were worse for the lower threshold in all countries but Zambia (*Supplementary file 2C*). Trip distances were defined as the haversine distance between centroids of districts. Shapefiles for the different countries were downloaded from DIVA-GIS (https://www.diva-gis.org/). Generally,

the district identification numbers were assigned by the map source, with district IDs being clustered within their respective region.

## Mobile phone data

Anonymized call data records (CDRs) were provided by the leading mobile phone provider in each country (*Supplementary file 1A*). Two districts in Namibia did not have data (Oshakati and Uuvud-hiya) and were excluded from analysis. The Burkina Faso provider shared CDRs from a subset of randomly selected subscribers (100,000, ~ 1.4 % of subscribers), as opposed to the other coun-tries' providers that shared CDRs from all of their subscribers. The duration of and year(s) covered by the CDR datasets varied by country, defined by different data sharing agreements: Namibia's ran October 2, 2010 – April 30, 2014; Kenya's ran June 1, 2008 – July 3, 2009 (excluding the month of February, 2009); Burkina Faso's ran January 1, 2016 – December 31, 2016; and Zambia's ran August 1, 2020 – December 30, 2020. Briefly, CDRs for each country were first aggregated to tower locations and then to districts for each country. A similar method described elsewhere (*Zu Erbach-Schoenberg et al., 2016*) was used to assign cell towers to districts. Briefly, if a cell tower's coverage zone fell squarely within one district, all CDRs associated with that tower were assigned to that district. If the coverage zone spanned more than one district, the number of CDRs assigned to each district was split according to the area of overlap between the coverage zone and districts. We only considered travel that crossed district boundaries, not local movement within the district. The average number of total monthly trips taken between each origin and destination was calcu-lated and the proportion of trips was calculated for each origin by normalizing the trip counts to a given destination by the total trips made from that origin. Trip types were defined by origin and destination, either by urbanicity (urban or rural) or by region (intra- or inter-regional). Statistical and spatial analysis was done in R v3.6.3.

## Mobility models

We compared the ability of eight variations of the gravity model and a basic radiation model to capture the heterogeneity in trip counts ($T_{ij}$) between each origin ($i$) and destination ($j$).

## Gravity models

The gravity model estimates the trip counts, $T_{i,j}$, as a function of the population sizes at the origin ($P_i$) and destination ($P_j$) and deterrence function that depends on the distance between the two locations ($d_{i,j}$) (Equation 1).

$$\hat{T}_{i,j} = \theta \frac{P_i^\alpha P_j^\beta}{f(d_{i,j})} \tag{1}$$

Here, $\alpha$ and $\beta$ are non-negative parameters that scale the strength of association between $i$ and $j$; $\theta$ acts as a proportionality constant, and $f(d_{ij})$ is the penalty associated with a trip distance (d, in kilometers). Both the power $\left(f(d_{ij}) = d_{ij}^\gamma\right)$ and exponential $\left(f(d_{ij}) = exp\left(\frac{d_{i,j}}{D}\right)\right)$ forms of the deterrence function were tested, where $\gamma$ is a non-negative parameter that determines the rate at which the number of trips decays with trip distance and D is a non-negative parameter that captures the deter-rence distance (*Chen, 2015*). The number of trips increases with larger values of $\alpha$ and $\beta$ and smaller values of $\gamma$ or D. We tested eight model variations of Equation 1 in which these parameters were allowed to vary according to aspects of the origin and destination.

Power variants:

Basic: parameters are fitted to the full set of trips,

$$\hat{T}_{i,j} = \theta \frac{P_i^\alpha P_j^\beta}{d_{i,j}^\gamma} \tag{2}$$

Urbanicity: parameters are fitted to trips categorized by the urbanicity of the origin and destination (k = 1 : 4 for rural-rural, rural-urban, urban-rural, and urban-urban),

$$\hat{T}_{i,j} = \theta \frac{P_i^{\alpha_k} P_j^{\beta_k}}{d_{i,j}^{\gamma_k}} \begin{cases} k = 1 \ \textit{if urbanicity}_i = rural \wedge urbanicity_j = rural \\ k = 2 \ \textit{if urbanicity}_i = rural \wedge urbanicity_j = urban \\ k = 3 \ \textit{if urbanicity}_i = urban \wedge urbanicity_j = rural \\ k = 4 \ \textit{if urbanicity}_i = urban \wedge urbanicity_j = urban \end{cases} \quad (3)$$

Regional: parameters are fitted based on trips categorized by whether the origin and destination of a trip were both in the same region (intra-regional) or in different regions (inter-regional) (m = 1 : 2),

$$\hat{T}_{i,j} = \theta \frac{P_i^{\alpha_m} P_j^{\beta_m}}{d_{i,j}^{\gamma_m}} \begin{cases} m = 1 \ \textit{if region}_i = region_j \\ m = 2 \ \textit{if region}_i \neq region_j \end{cases} \quad (4)$$

Regional-Urbanicity: parameters are fitted to trips categorized by both the region and urbanicity of the origin and destination (n = 1 : 8 for intra-regional- rural-to-rural, inter-regional-rural-to-rural, etc.).

$$\hat{T}_{i,j} = \theta \frac{P_i^{\alpha_n} P_j^{\beta_n}}{d_{i,j}^{\gamma_n}} \begin{cases} n = 1 \ \textit{if region}_i = region_j \wedge urbanicity_i = rural \wedge urbanicity_j = rural \\ n = 2 \ \textit{if region}_i \neq region_j \wedge urbanicity_i = rural \wedge urbanicity_j = urban \\ n = 3 \ \textit{if region}_i = region_j \wedge urbanicity_i = urban \wedge urbanicity_j = rural \\ n = 4 \ \textit{if region}_i \neq region_j \wedge urbanicity_i = urban \wedge urbanicity_j = urban \\ n = 5 \ \textit{if region}_i = region_j \wedge urbanicity_i = rural \wedge urbanicity_j = rural \\ n = 6 \ \textit{if region}_i \neq region_j \wedge urbanicity_i = rural \wedge urbanicity_j = urban \\ n = 7 \ \textit{if region}_i = region_j \wedge urbanicity_i = urban \wedge urbanicity_j = rural \\ n = 8 \ \textit{if region}_i \neq region_j \wedge urbanicity_i = urban \wedge urbanicity_j = urban \end{cases} \quad (5)$$

Exponential variants:

Basic: parameters are fitted to the full set of trips,

$$\hat{T}_{i,j} = \theta \frac{P_i^{\alpha} P_j^{\beta}}{exp\left(\frac{d_{i,j}}{D}\right)} \quad (6)$$

Urbanicity: parameters are fitted to trips categorized by the urbanicity of the origin and destination (k = 1 : 4 for rural – rural, rural – urban, urban – rural, and urban – urban. See definitions in Equation 3),

$$\hat{T}_{i,j} = \theta \frac{P_i^{\alpha_k} P_j^{\beta_k}}{exp\left(\frac{d_{i,j}}{D_k}\right)} \quad (7)$$

Regional: parameters are fitted based on trips categorized by whether the origin and destination of a trip were both in the same region (intra-regional) or in different regions (inter-regional) (m = 1 : 2. See definitions in Equation 4),

$$\hat{T}_{i,j} = \theta \frac{P_i^{\alpha_m} P_j^{\beta_m}}{exp\left(\frac{d_{i,j}}{D_m}\right)} \quad (8)$$

Regional-Urbanicity: parameters are fitted to trips categorized by both the region and urbanicity of the origin and destination (n = 1 : 8 for intra-regional- rural-to-rural, inter-regional-rural-to-rural, etc. See definitions in Equation 5),

$$\hat{T}_{i,j} = \theta \frac{P_i^{\alpha_n} P_j^{\beta_n}}{exp\left(\frac{d_{i,j}}{D_n}\right)} \quad (9)$$

We fit the gravity model parameters $\theta, \alpha, \beta, \gamma,$ and $D$ to observed trip counts extracted from mobile phone data ($m_{i,j}$) using Bayesian inference, where the model likelihood was assumed to have Poisson

error structure (Equation 10) and parameters were given uninformative Gamma priors. The gravity models were fitted to call data records using the R package 'mobility', which employs the JAGS (Just Another Gibbs Sampler) Bayesian MCMC algorithm and 'rjags' R package (found at https://github.com/COVID-19-Mobility-Data-Network/mobility, *John, 2021*). The posterior parameter estimates were then used to simulate human mobility patterns.

$$m_{i,j} \sim Pois(\hat{T}_{i,j}) \tag{10}$$

## Radiation model

Like the gravity model, the radiation model estimates a trip count, $T_{i,j}$, as a function of origin and destination population size (Equation 11); however, it differs in the way that it assumes that the probability of making a trip is also influenced by nearby potential destinations (*Simini et al., 2012*). Thus, $T_{i,j}$ is also dependent on the total population in the circle ($s_{i,j}$) centered at $i$ with a radius equal to $d_{i,j}$, excluding populations in $i$ and $j$, and we defined this value by summing the population sizes of districts that fell completely or partially within the radius. The number of trips emanating from origin $i$ is $T_i = \sigma P_i$, where $\sigma$ is the proportion of the entire country's population that traveled over a given time period ($P = \sum_i P_i$). We fit the parameters $\sigma$ associated with each $T_{i,j}$ to trips calculated from mobile phone data ($m_{i,j}$) using a Poisson error structure (Equation 10). While additional forms of the radiation model have been explored elsewhere (*Bjørnstad et al., 2019*), we focused on a form that normalizes $T_{i,j}$ for a finite system (*Masucci et al., 2013*).

$$\hat{T}_{i,j} = \frac{\sigma P_i}{1 - \frac{P_i}{P}} \frac{P_i P_j}{(P_i + s_{i,j})(P_i + P_j + s_{i,j})} \tag{11}$$

## Model comparisons

Models were compared using the Deviance Information Criterion (DIC), a criterion designed for MCMC outputs that assesses a model's trade-off between goodness of fit and complexity (*Shriner and Yi, 2009*; *Spiegelhalter et al., 2002*). Models fit to the same datasets (e.g., from the same country) were compared and those with the lowest DIC were selected as the best model. To determine the distribution of trips that were over- or underestimated for a given model, the ratio of estimated to observed trip counts for each route was calculated. Given that the distribution of ratios ranged nine orders of magnitude, the general accuracy of model estimates for specific trip types was evaluated by comparing the proportion of trips with model estimates that fell within ±10 % of the observed trips.

## Modifiable areal unit problem

To explore the impact of modifiable areal unit problem (MAUP) (e.g., the effect of the arbitrary definition of administrative units the spatial distribution of CDRs and population), we fit and compared models for a range of administrative units. We reran the gravity models with a power decay function at the administrative one unit (region) level for all countries and administrative three unit level for Burkina Faso, the only country whose dataset was supplied at the administrative three unit level. Note that the models involving regionality could not be run at the administrative one unit. Regardless of the administrative unit level used (e.g., smallest or largest district sizes), the general trend in model ranking was preserved (*Supplementary file 2B*).

## Data availability

A different form of the datasets from Kenya and Namibia that were negotiated in a prior negation are available as supplements of (*Ruktanonchai et al., 2016* and *Wesolowski et al., 2015b*). Individuals interested in the dataset from Zambia may contact the authors with requests.

## Acknowledgements

Research reported in this publication was supported in part by the National Library Of Medicine of the National Institutes of Health under Award Number DP2LM013102 (HRM, JRG, APW) and 1R01AI160780-01 (APW), a Career Award at the Scientific Interface by the Burroughs Wellcome Fund (HRM, JRG, APW), the Swiss Agency for Development and Cooperation - 2iE partie scientifique

– through "Projet 3E Afrique, Burkina Faso" (JSP, TM, AR), and a Swiss National Science Foundation grant under Award Number 200021–172578 (JSP, AR). The content is solely the responsibility of the authors and does not necessarily represent the official views of the funders.

## Additional information

### Competing interests
Amy Wesolowski: Reviewing editor, eLife. The other authors declare that no competing interests exist.

### Funding

| Funder | Grant reference number | Author |
| --- | --- | --- |
| National Institutes of Health | DP2LM013102 | Amy Wesolowski |
| Burroughs Wellcome Fund | | Amy Wesolowski |
| Swiss Agency for Development and Cooperation | | Andrea Rinaldo |
| Swiss National Science Foundation | 200021-172578 | Andrea Rinaldo |
| National Institutes of Health | 1R01AI160780-01 | Amy Wesolowski |

The funders had no role in study design, data collection and interpretation, or the decision to submit the work for publication.

### Author contributions
Hannah R Meredith, Conceptualization, Formal analysis, Investigation, Methodology, Writing - original draft, Writing – review and editing; John R Giles, Software, Writing – review and editing; Javier Perez-Saez, Data curation, Validation, Writing – review and editing; Théophile Mande, Caroline O Buckee, Resources; Andrea Rinaldo, Data curation, Funding acquisition, Resources, Writing – review and editing; Simon Mutembo, Elliot N Kabalo, Kabondo Makungo, Data curation; Andrew J Tatem, C Jessica E Metcalf, Resources, Writing – review and editing; Amy Wesolowski, Conceptualization, Data curation, Funding acquisition, Resources, Supervision, Writing – review and editing

### Author ORCIDs
Hannah R Meredith ![ORCID] http://orcid.org/0000-0002-5315-7568
Caroline O Buckee ![ORCID] http://orcid.org/0000-0002-8386-5899
C Jessica E Metcalf ![ORCID] http://orcid.org/0000-0003-3166-7521
Amy Wesolowski ![ORCID] http://orcid.org/0000-0001-6320-3575

### Decision letter and Author response
Decision letter https://doi.org/10.7554/eLife.68441.sa1
Author response https://doi.org/10.7554/eLife.68441.sa2

## Additional files

### Supplementary files
• Supplementary file 1. Country and trip details. A. Basic characteristics of countries and trips. B. Key of region and district IDs and names. See *Figure 1—figure supplement 9* for a map.

• Supplementary file 2. Further analysis of model fits. A. Gravity model variations and radiation model, ranked for each country based on Deviance Information Criterion (DIC) (Standard Deviation) and percent change (%Δ) from basic model. A similar trend in ranking models by best fit was seen for gravity models with power or exponential distance kernel. B. Model fits ranked by Deviance Information Criterion (DIC) for each country. Although the different definitions of urbanicity impacted the distribution of urbanicity trip types, the overall ranking of the model fits was not

affected. Generally, the models that used the lower urban threshold (10 % urban grid cells) had larger DICs (worse fits) than the models that used the higher urban threshold (50 % urban grid cells). The gravity models with the power distance kernel were used here. C. Model fits ranked by Deviance Information Criterion (DIC) for each country at administrative levels 1–3 (when available). Although the different administrative unit boundaries impacted the size of the DIC, the overall ranking of the model fits was not affected. Generally, the models that the larger administrative units (administrative one units) had smaller DICs (better fits) than the models that used the smaller administrative units (administrative three units).

• Supplementary file 3. Proportion of trips estimated within target interval. A. For each trip type and country, the model was reported that estimated the highest proportion of trips with estimated trip counts that fell within ±10 %the observed trips (% trips). In situations where the proportion of trips of estimated by two models differed by less than 1%, both models were included. The distance kernel used is indicated by exp (exponential) or pwr (power). See B and C for the trip proportions for all models. B. For each trip type in each country, the percentage of estimated trips that fell within the selected interval of ±10 % of the observed trip count. (Power distance kernel used in gravity models) C. For each trip type in each country, the percentage of estimated trips that fell within the selected interval of ±10 % of the observed trip count. (Exponential distance kernel used in gravity models).

• Transparent reporting form

### Data availability

Due to data sharing agreements with the mobile phone companies, the call data records used in this study are not directly available. However, the Burkina Faso data sharing agreement allows for a jittered dataset of monthly aggregated trips between administrative 2 units, which can be found on Dryad as "Burkina Faso mobility data with some noise" (https://doi.org/10.5061/dryad.fn2z34tt6). A different form of the datasets from Kenya and Namibia that were negotiated in a prior negation are available as supplements of (Ruktanonchai et al., 2016) and (Wesolowski et al., 2015b). Individuals interested in the dataset from Zambia may contact the authors with requests. The code used to analyze the mobile phone data and run the models can be found on github at https://github.com/hrmeredith12/Rural-mobility-models.git (copy archived at https://archive.softwareheritage.org/swh:1:rev:cfc77221c574dad23c3204cd6c5d5fadcb1ce385).

The following dataset was generated:

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
