## [Decision Letter]

**Acceptance summary:**

This paper presents a comparison of several mathematical models of human mobility in low and middle income settings, fitted to mobile phone usage data. This article is of particular interest to researchers within the field of human mobility studies, in addition it is also of potential interest to a broader audience with interests in the application of human movement patterns such as the spread of infectious diseases, health service access and utilization, logistics and more.

**Decision letter after peer review:**

Thank you for submitting your article "Characterizing human mobility patterns in rural settings of Sub-Saharan Africa" for consideration by *eLife*. Your article has been reviewed by 3 peer reviewers, including Jennifer Flegg as the Reviewing Editor and Reviewer #1, and the evaluation has been overseen by Aleksandra Walczak as the Senior Editor. The following individual involved in review of your submission has agreed to reveal their identity: Francois Rerolle (Reviewer #3).

Essential revisions:

1) Is there data across more countries in Africa? If so, it would be great to see the applicability of the recommendations of models to similar settings.

2) Can the authors be clearer about how they select which model was most appropriate for different contexts eg the interplay between model fit and model complexity?

3) Can the authors comment on the likely suitability of the recommendations outside of Africa?

4) Why were only 1.4% of randomly selected subscribers from the Burkina Faso provider included? Was this the only data provided to authors or was this the percentage of subscribers left once the authors excluded local movement within the district?

5) Line 138 of the manuscript reads "By comparing and evaluating a wide range of models…" This statement is too broad and potentially misleading given that all models (except the radiation model) utilised in this study are variations of the gravity model.

6) Have the authors considered deriving an ensemble model as part of their future work? If they were to pursue this idea, it would be interesting to see the results from other interaction models first such as an intervening opportunities model.

7) It would also be great to see something similar done for LMICs on a different continent, although I note this is beyond the scope of the current manuscript.

8) Given the differences in time frame for call data records (provided in the supporting information) used for each country, this should be discussed in the Materials and methods section of the manuscript.

9) The rationale behind the choice of the target interval of 0.5-2 times the observed trip counts determined to be appropriate for the percentage of estimated trips should be included.

10) From the title, I was expecting a bit more description/characterization of what the human travel looks like on these settings. I believe the article needs to describe the overall patterns of human mobility highlighted in the data collected rather than focus too much on model performance. How many trips are there per inhabitants per year? How much does it depend on distance (simple interpretation of decay function: trips decreased by X% every 100km or something like that), population density (there are Y times as many trips to urban areas compared to rural areas). And how it varied across the 3 countries.

11) The analysis ends up suggesting a variation of basic models with more parameters, adapted to the rural settings of sub-saharan Africa. Shouldn't the introduction provide more background on the performances of the basic models and their more parametrized variations in settings where they have been developed (high-income countries)? One would expect the basic model not to perform as well across all mobility settings of high-income countries. Similar adjustment for regionality and urbanicity may be needed (and previously evaluated) in more studied settings of rural America for instance?

12) Figure 1 J-L: For those type of plots, it would be worth spending a sentence or two to describe interpretations. For the axis, I know there is a reference to supplementary materials but it took me a while to understand that these were just numbered locations? Same comment with respect to the coloring, what proportion is that referencing to? Within a destination? Overall? Why categorize the coloring and not use a continuous nuanced color palette? Breaking by log values seems arbitral and the bluest categories contains a very wide range of proportion (0.1 to 1). Also, have you considered sorting the locations by population density? It might convincingly demonstrate the limitations of the basic model.

13) The introduction mentions other studies in LMIC that have adjusted basic models with individual levels factors such as education, SES, gender,etc and improved the fit. In this study, the authors propose a different and higher-level type of adjustment (regionality and urbanicity of trips' origin/destination). Are the authors also able to adjust for those individual level factors or are they absent from the mobile phone data? The comparison with previous work and improvements suggested in the article would be more convincing if both type of adjustment are done and combined on the same datasets. Otherwise we can't really compare the 2 approaches and advocate in favor of one or the other, can we?

14) Stratifying models by features (urbanicity and regionality) limits generalizability to other settings important features and/or where cut points (e.g, between rural and urban) may need to be different. Thanks to Bayesian analysis, have the authors considered modeling parameters of the gravity models as continuous functions of population density instead? Although it would decrease interpretability of the results, it would improve generalizability of the work and potentially result in significant fit improvements.

15) The authors compare models' performance based on % change in DIC. I am not as familiar with DIC, but I thought absolute changes (for AIC) were more relevant. Can the authors please clarify?

16) Lines 275: The selected interval (1/2; 2) seems both arbitral and pretty wide. Could the authors elaborate a bit more on it?

17) Figure 3B: Out of the gravity models, the bell curve for the basic model seems to be the closest to the 1:1 ratio except for the rural-rural trips. Doesn't this mean it is well performing?

18) At first, it is a bit confusing to use similar notation for functions used in the equations to denote exponential/power decay f() and stratification of parameters f(all trips), f(urbanicity),…. Can the authors please revise the notation?

---

## [Author Response]

Essential revisions:1) Is there data across more countries in Africa? If so, it would be great to see the applicability of the recommendations of models to similar settings.

While the manuscript was under review, we gained access to a new mobile phone dataset from Zambia. We have updated the analysis to include this additional country. While data sharing was not part of the original agreement we have with the regulator, we are working with them to make some of the aggregated data available. In the meantime, individuals may contact us with data requests.

One of the biggest limitations of mobile phone data is its accessibility, and we are unable to include data from more countries across Africa at this time. A few data sets, including those from Senegal (https://doi.org/10.4081/gh.2016.408) and Cote d’Ivoire (https://doi.org/10.1038/srep02923), have been published, we did not currently have access to these data for this work.

2) Can the authors be clearer about how they select which model was most appropriate for different contexts eg the interplay between model fit and model complexity?

The model that best predicted all trips was selected based on the Deviance Information Criterion (DIC). This criterion incorporates the trade-off between model fit and model complexity. We have updated the text to provide a clearer description of what the DIC is and how it should be interpreted:

“Models were compared using the Deviance Information Criterion (DIC), a criterion designed for MCMC outputs that assesses a model’s trade-off between goodness of fit and complexity (Shriner and Yi, 2009; Spiegelhalter et al., 2002). Models fit to the same datasets (e.g., from the same country) were compared and those with the lowest DIC were selected as the best model.” (Lines 309-312)

The DIC is best for assessing how well the model fit the entire dataset, not specific trip contexts (e.g., Rural-rural or inter-regional). To evaluate a model in a specific context, we focused on predictive accuracy. Oftentimes metrics such as mean squared error are used to assess model accuracy. However, we wanted to be able to better illustrate not just the accuracy of the estimate, but also if these estimates were likely to be over- or under- estimated for a particular trip. Thus, we used the ratio of modeled:observed trips to evaluate model accuracy for each origin-destination pair. (Lines 312-316)

3) Can the authors comment on the likely suitability of the recommendations outside of Africa?

While this remains to be studied, we surmise that our findings will likely apply to similar settings outside of Africa. Differences geographically and more urban settings (as opposed to many settings in Sub-Saharan Africa) may change which model is the best fitting, however we believe this approach could be used more generally to investigate mobility models. We have incorporated this point in the discussion.

“Future work could also evaluate the models tested here in other countries, both within and outside of Sub-Saharan Africa. The increasing availability of mobility data has enabled the comparison of global human mobility patterns and revealed that (a) longer distance trip (> 20 km) patterns are similar across low- and high-income settings (Kraemer et al., 2020) and (b) that mobility patterns in rural (sparsely populated) areas differ from those in urban (densely populated) areas (Liu et al., 2015). This suggests that our findings based on trips aggregated to the region or district level will be generalizable to other countries from a range of income settings. However, mobility patterns at smaller spatial scales appear to be different for low- and high-income settings (Kraemer et al., 2020), and requires further investigation.” (Lines 177-186)

4) Why were only 1.4% of randomly selected subscribers from the Burkina Faso provider included? Was this the only data provided to authors or was this the percentage of subscribers left once the authors excluded local movement within the district?

Telecel only provided a subset of their subscribers for analysis. We have clarified the methods to better reflect this limitation.

“The Burkina Faso provider shared CDRs from a subset of randomly selected subscribers (100,000, ~1.4% of subscribers), as opposed to the other countries’ providers that shared CDRs from all of their subscribers” (Lines 237-239)

5) Line 138 of the manuscript reads "By comparing and evaluating a wide range of models…" This statement is too broad and potentially misleading given that all models (except the radiation model) utilised in this study are variations of the gravity model.

We have updated the statement to: “By comparing and evaluating a range of gravity models…” (Line 155)

6) Have the authors considered deriving an ensemble model as part of their future work? If they were to pursue this idea, it would be interesting to see the results from other interaction models first such as an intervening opportunities model.

The reviewer makes a good point. An ensemble model would be an interesting approach to explore in future studies. A recent study has shown that an ensemble model has improved estimates of mobility patterns in Australia. We have included this point in the discussion:

“Moving forward, the need to select a single model could be mitigated and estimates of mobility patterns could be improved by developing an ensemble model. Recently, an ensemble model outperformed individual models in estimating human mobility patterns in Australia, combining different mobility models as well as data types to optimize mobility estimates across different spatial scales (McCulloch et al., 2021).” (Lines 172-175)

7) It would also be great to see something similar done for LMICs on a different continent, although I note this is beyond the scope of the current manuscript.

We thank the reviewer for the suggestion. Indeed, this would be interesting, but is outside the scope of this study. We have addressed this in our response to comment #3 above and are also making our code available on github to facilitate testing datasets from other settings.

8) Given the differences in time frame for call data records (provided in the supporting information) used for each country, this should be discussed in the Materials and methods section of the manuscript.

We have updated the methods section to include the time frames for the CDRs:

“The duration of and year(s) covered by the CDR datasets varied by country, defined by relevant data sharing agreements: Namibia’s ran October 2, 2010 – April 30, 2014; Kenya’s ran June 1, 2008 – July 3, 2009 (excluding the month of February, 2009); Burkina Faso’s ran January 1, 2016 – December 31, 2016; and Zambia’s ran August 1, 2020 – December 30, 2020.” (Lines 239-242)

9) The rationale behind the choice of the target interval of 0.5-2 times the observed trip counts determined to be appropriate for the percentage of estimated trips should be included.

To assess the accuracy of each model’s estimates, we considered how much each model over- or under-estimated trips of a given category. Given that the ratio of predicted to observed trip counts ranged orders of magnitude (1E-03 to 1E+06), we selected an interval that would represent a reasonable degree of over/under-estimation. In response to a comment below, we have reduced the target interval to be ± 10% the observed trip counts.

“To determine the distribution of trips that were over- or under-estimated for a given model, the ratio of estimated to observed trip counts for each route was calculated. Given that the distribution of ratios ranged nine orders of magnitude, the general accuracy of model estimates for specific trip types was evaluated by comparing the proportion of trips with model estimates that fell within ±10% of the observed trips.” (Lines 312-316)

10) From the title, I was expecting a bit more description/characterization of what the human travel looks like on these settings. I believe the article needs to describe the overall patterns of human mobility highlighted in the data collected rather than focus too much on model performance. How many trips are there per inhabitants per year? How much does it depend on distance (simple interpretation of decay function: trips decreased by X% every 100km or something like that), population density (there are Y times as many trips to urban areas compared to rural areas). And how it varied across the 3 countries.

We thank the reviewer for the suggestion. While the call data records are aggregated spatially such that individual level trends cannot be analyzed, we have expanded upon the population level description of the trips. For instance, the distribution of trips between rural and/or urban locations and within or between regions differed for each country.

“Trips were concentrated between districts within the same region (administrative level 1) to varying degrees (30% in Burkina Faso, 45% in Kenya, 62% in Namibia, and 72% in Zambia-62% of trips, depending on the country) and to a few common destinations, including the district where the capital was located (Figure 1JI-L, Supplementary file 1A). Although Namibia, Burkina Faso, and Zambia each consisted of ~95% predominantly rural districts, the distribution of monthly trips between urban and rural districts varied across countries. The majority of Namibia’s and Burkina Faso’s trips were between rural locations (62% and 70.5% of all trips, respectively), while Zambia’s trips were split between rural locations (53%) or rural and urban locations (46%). Kenya, with 56% predominantly urban districts, had the largest proportion of monthly trips between urban locations (70%).” (Lines 94-102)

11) The analysis ends up suggesting a variation of basic models with more parameters, adapted to the rural settings of sub-saharan Africa. Shouldn't the introduction provide more background on the performances of the basic models and their more parametrized variations in settings where they have been developed (high-income countries)? One would expect the basic model not to perform as well across all mobility settings of high-income countries. Similar adjustment for regionality and urbanicity may be needed (and previously evaluated) in more studied settings of rural America for instance?

We thank the reviewers for the chance to provide more background on how the gravity model performs in high income settings. We have updated the introduction with the following:

“In high income settings, the standard gravity model has been shown to perform well when predicting commuter movement between cities (Masucci et al., 2013) and perform poorly when predicting movement across areas with heterogeneity in demographics and population density or in rural areas (Truscott and Ferguson, 2012; Xia et al., 2004).” (Lines 57-60)

We have also mentioned in the discussion that these model adjustments may also help improve trip estimates in high income countries.

“Model estimates of travel patterns in high income countries may also benefit from accounting for urbanicity and regionality (Truscott and Ferguson, 2012; Xia et al., 2004).” (Lines 178-180)

12) Figure 1 J-L: For those type of plots, it would be worth spending a sentence or two to describe interpretations. For the axis, I know there is a reference to supplementary materials but it took me a while to understand that these were just numbered locations? Same comment with respect to the coloring, what proportion is that referencing to? Within a destination? Overall? Why categorize the coloring and not use a continuous nuanced color palette? Breaking by log values seems arbitral and the bluest categories contains a very wide range of proportion (0.1 to 1). Also, have you considered sorting the locations by population density? It might convincingly demonstrate the limitations of the basic model.

We thank the reviewers for the opportunity to clarify the OD matrices. We have updated the legend to further explain the coloring scheme and axis tics.

“The columns and rows of the OD matrix are ordered by district ID, the order of which is typically assigned by shapefiles or mobile phone operators. The capital district is indicated by the black arrow on the x- and y-axes. The colors indicate the proportion of an origin’s trips made to each destination (with light blue representing destinations visited infrequently and dark blue representing destinations visited most frequently). ” (Figure 1 caption, Lines 365-369)

The trip colors were categorized into bins to help better visualize patterns. Relatively few destinations made up > 10% of an origin’s trips, which is lost when using a continuous color scale. We also believe it is easier to interpret the plot with fewer colors (e.g., uncommon destination: < 0. 1% of all trips <-> common destination: 10-100% of all trips)

We considered sorting by urbanicity (a proxy for density; Author response image 1 middle panel) and population size (right panel); however, the clearest trends were observed when sorting by district ID (left).

**Author response image 1. sa2fig1:** 

13) The introduction mentions other studies in LMIC that have adjusted basic models with individual levels factors such as education, SES, gender,etc and improved the fit. In this study, the authors propose a different and higher-level type of adjustment (regionality and urbanicity of trips' origin/destination). Are the authors also able to adjust for those individual level factors or are they absent from the mobile phone data? The comparison with previous work and improvements suggested in the article would be more convincing if both type of adjustment are done and combined on the same datasets. Otherwise we can't really compare the 2 approaches and advocate in favor of one or the other, can we?

While it would be interesting to see how accounting for urbanicity and regionality impacts the fit of models with individual level factors, our datasets were de-identified by the mobile phone operators and aggregated to a spatial and temporal level such that we could not analyze individual level factors. To investigate this in the future, perhaps different datasets, such as travel surveys, census data, or data collected from mobile phone apps, could be incorporated into an ensemble model. We have incorporated this into the discussion:

“Using this approach, additional individual level information (e.g., gender, age, occupation) collected from other data sources could be incorporated to study their impact on model fit.” (Lines 175-177)

14) Stratifying models by features (urbanicity and regionality) limits generalizability to other settings important features and/or where cut points (e.g, between rural and urban) may need to be different. Thanks to Bayesian analysis, have the authors considered odelling parameters of the gravity models as continuous functions of population density instead? Although it would decrease interpretability of the results, it would improve generalizability of the work and potentially result in significant fit improvements.

We appreciate the reviewer’s suggestion to make the models more generalizable. Population density can be approximated by urbanicity (proportion of the administrative unit that falls within a region with a population density above a standard/generalizable threshold established by WorldPop). As a test, we considered fitting the parameters as functions of population density for Namibia, however the model fit was not significantly improved (see Author response table 1). Furthermore, fitting by population density/urbanicity did not allow for the parameters to vary as much as the regional-urbanicity model, making the model less flexible. As such, we decided not to run this model variation for the other countries and did not include it in the manuscript.

**Author response table 1. sa2table1:** 

	DIC	γ	α	β
Basic Model (pwr)	6.12E+06	1.42	1.17	1.17
Regional-Urbanicity Model (pwr)	3.62E+06	0.83:5.35	0.68:1.88	0.68:1.06
New population density model (pwr)	4.97 E+06	0.59:3.11	0.11:1.93	0.45:1.64

15) The authors compare models' performance based on % change in DIC. I am not as familiar with DIC, but I thought absolute changes (for AIC) were more relevant. Can the authors please clarify?

Both the AIC and DIC aim at balancing goodness of model fit to data and model complexity and are used in the same way to compare and select “best” models. The difference is that the DIC estimates model complexity as a function of the posterior in a Bayesian setup, whereas the AIC is computed as a function of the maximum likelihood estimate. Thus, calculating the AIC is more appropriate for approaches based on maximum likelihood whereas the DIC is more appropriate for approaches based on MCMC. In response to reviewer comment #2, we have expanded the text to clarify our choice of DIC for model selection and how it should be interpreted.

16) Lines 275: The selected interval (1/2; 2) seems both arbitral and pretty wide. Could the authors elaborate a bit more on it?

We thank the reviewer for their comment. We have reduced the selected interval to ± 10% the observed trip counts and have further elaborated on its selection. Please see our full response answer to the similar question/comment (#9) above.

17) Figure 3B: Out of the gravity models, the bell curve for the basic model seems to be the closest to the 1:1 ratio except for the rural-rural trips. Doesn't this mean it is well performing?

The reviewer’s interpretation is correct – the closer the distribution’s mean is to 1 and the tighter the spread, the better that model is at predicting those trips. However, in Figure 3B, the only trip type the Basic Model is best at predicting is the Urban-Urban trips. For Rural-Urban and Urban-Rural trips, the Regional model’s mean was closer to 1 than the Basic model’s. For Rural-Rural trips, the Radiation model’s mean was closer to 1 than the Basic model’s. We have changed the lines indicating the 1:1 ratio as well as the boundaries of the selected interval (now white) to be more distinguishable from the line of the mean ratio (black).

18) At first, it is a bit confusing to use similar notation for functions used in the equations to denote exponential/power decay f() and stratification of parameters f(all trips), f(urbanicity),…. Can the authors please revise the notation?

The notation has been revised to the following (see Lines 272-278):T^i,j=θPiαkPjβkdi,jγk{k=1 if urbanicityi=ruralAND urbanicityj=rural k=2 if urbanicityi=rural AND urbanicityj=urbank=3 if urbanicityi=urban ANDurbanicityj=ruralk=4 if urbanicityi=urban AND urbanicityj=urban